# Counterfactual Fairness in Synthetic Data Generation

**Mahed Abroshan**
The Alan Turing Institute, London, UK
`mabroshan@turing.ac.uk`

**Mohammad Mahdi Khalili**[*]
Yahoo! Research, NYC, NY, USA
`mahdi.khalili@yahooinc.com`

**Andrew Elliott**[†]
Department of Mathematics and Statistics
University of Glasgow, Glasgow, Scotland
`andrew.elliott@glasgow.ac.uk`

## Abstract

Synthetic data generation (SDG) is proposed as a promising solution for data sharing as in many high-stake applications due to privacy concerns, releasing the real dataset is not an option. While the main goal of private SDG is to create a dataset that preserves the privacy of individuals contributing to the dataset, the use of synthetic data also creates an opportunity to improve the fairness issue at the source. Since there exist historical biases in the datasets, using the biased data to train an ML model can lead to an unfair model which may exacerbate the discrimination. Using synthetic data, we can attempt to remove the bias from the dataset before releasing the data. In this work, we formalize the definition of fairness in synthetic data generation and propose a method to achieve counterfactual fairness.

## 1 Introduction

Data sharing has become a bottleneck for the progress of machine learning methods. In most important applications of machine learning like healthcare and finance, privacy concerns and existing rules do not allow data holders to release the data. Differentially private synthetically generated data can be a promising solution for the data sharing problem [1, 2]. These models try to create a dataset that resembles the real data, while also satisfying some level of differential privacy [3]. The criteria for comparing these models are usually fidelity, diversity, and privacy. The first two criteria account for how similar is the generated data to the real data and the third one measures the privacy leakage of the model.

Another relevant concern which can be an obstacle to the use of ML is the inherent fairness or lack thereof in data sets. It has been shown that the ML models can reflect and in some cases exacerbate the existing biases in the dataset [4, 5, 6, 7]. Several fairness notions and methods to achieve these notions have been proposed in the literature [8]. These methods impose an additional constraint (depending on the fairness notion of choice) on the predictor to ensure that it complies with the fairness criterion. It is, however, a task for the user of the data to ensure the fairness constraint is satisfied. Using synthetic data creates an interesting opportunity to make sure that the released dataset is fair. Such that when it is used in a downstream task, the resulting model will be also fair. The goal of fair synthetic data generation is to generate a dataset that is as close as possible to the real data while removing the discriminatory biases existing in the data. Inevitably, using this method will alter the distribution of the data and hence decreases the performance of the model trained on the generated data. However, we note attempting to train a fair predictor on real data (rather than

---

[*]The author is also with the Ohio State University, Columbus, Ohio.
[†]The author is also with the Alan Turing Institute, London, UK.

NeurIPS 2022 Workshop on Synthetic Data for Empowering ML Research.

synthetic), will also cause some performance drop when compared to the case where a fairness constraint is not imposed. There are several different notions introduced for fairness, choosing the right notion depends on the policy maker's preference and the task at hand. In this work, we consider the counterfactual fairness (CF) notion [9]. Our contribution is formalising the definition of fairness for synthetic data in Section 2.2, and a new method for achieving CF is proposed in Section 3.

## 1.1 Related work

The most relevant works to the problem we consider here are [10, 11, 12]. In [10], FairGAN, a GAN-based method is introduced to create a dataset that satisfies the statistical parity notion (see Section 2.1 for the definition of this notion). Their problem setting is slightly different from what we consider here. They want to create a dataset, such that every predictor trained on the dataset satisfies the fairness notion, whereas we want to align incentives of a data user and have a synthetic dataset such that an accurate predictor satisfies the fairness notion (see Appendix for a more detailed comparison). In [12], CFGAN is proposed to synthesise a dataset that satisfies CF. In Section 3, we show that their method is not satisfying CF, but a relevant notion defined in Definition 5. Finally, DECAF [11], is a casually-aware method which is proposed to create fair data for a few notions which does not include CF. We have a discussion on their method and the definition they propose in the appendix.

## 2 Fairness for synthetic data generation (SDG)

In this section, our goal is to provide a general definition of fairness in synthetic data generation. We also review the existing definitions in previous work [10, 11, 12] and compare them with our definition. We will first review some of the fairness notions that are defined for a given predictor.

### 2.1 Algorithmic fairness

Algorithmic fairness is a well-established area [13], with several fairness definitions for supervised models. The appropriate notion may be different for different datasets/tasks and for different policy makers. A subset of the most used definitions are as follows (more definitions can be found in [8]). Let $X$ denote the features, $A$ the sensitive attribute, and $Y$ the output in the dataset.

**Definition 1.** Statistical Parity (SP) [14], also known as demographic parity: A predictor $\hat{Y}$ satisfies demographic parity if $P(\hat{Y}|A = 0) = P(\hat{Y}|A = 1)$.

**Definition 2.** Equal Opportunity (EO) [13]: A binary predictor $\hat{Y}$ satisfies equal opportunity with respect to $A$ and $Y$ if $P(\hat{Y}|A = 0, Y = 1) = P(\hat{Y}|A = 1, Y = 1)$.

**Definition 3.** Counterfactual Fairness (CF) [9]: A predictor $\hat{Y}$ satisfies counterfactual fairness if for any context $A = a$ and $X = x$, $P(\hat{Y}_{A \leftarrow a} = y|X = x, A = a) = P(\hat{Y}_{A \leftarrow a'} = y|X = x, A = a)$ holds for all value of $y$ and $a' \in \mathcal{A}$.

The above fairness notions are all defined to assess the fairness of a given predictor. However, here we want to have a "fair dataset", and there is no predictor immediately in the picture, thus there is no variable $\hat{Y}$ which was used in all of the above definitions.

### 2.2 Fairness in SDG setting

The goal of fair SDG is that given a dataset $D = \{(X_i, A_i, Y_i)\}_{i=1}^{n}$ where $(X, A, Y) \sim P$, create a dataset $D'$ drawn from $(X, A, Y) \sim P'$ such that when $D'$ is used in a downstream task as the training dataset the resulting predictor satisfies some fairness notion of choice. Note that no matter what the data is, an end user can always create an unfair predictor (e.g., by only accepting men regardless of the rest of the features). Thus, the goal of fair SDG should be aligning incentives of the downstream user, such that maximizing the accuracy results in a fair classifier. Therefore, we want to argue that it is sensible to impose the fairness constraint on the labels of the generated data distribution $P'$ instead of the predicted label. For example, for demographic parity, we can impose that $P'(Y|A = 0) = P'(Y|A = 1)$. While this is doable for some of the definitions like

demographic parity, it is not possible for notions that depend explicitly on a downstream task like equal opportunity as has also been pointed out in [11].

Our definition of fair SDG can be formulated as a constrained optimization, where the goal is to find a distribution $P'(X, A, Y)$ such that it satisfies a certain fairness notion (adopting the notation in [11], we denote it by $\mathscr{I}((X, Y, A), P')$-fairness), while minimizing the distance $P'$ from the real data $P$, where $d$ is any distance of choice:

$$\min_{P'} d(P, P') \quad s.t. \quad \mathscr{I}((X, Y, A), P') - \text{fairness}. \tag{1}$$

To show that this definition is helpful, we prove that when an accurate predictor is trained on the fair dataset $P'$, then this predictor will satisfy some level of fairness (Proposition 1). Further, we will be using this predictor on the actual data which will be drawn from $P$. Therefore, we show that the predictor will have a reasonable accuracy on $P$ too (Proposition 2), and it will satisfy some level of fairness w.r.t $P$ (Proposition 3). We prove the following propositions for Statistical Parity (SP) definition, proving them for other fairness notions could be a direction for future work. In the results below we assume that $\mathcal{X}$ is a countable set. The proofs of these propositions and formal definition of $\mathbb{TV}$ distance are given in the appendix.

**Proposition 1.** If a distribution $P'$ approximately satisfies SP, that is $\mathbb{TV}(P'(Y|A = 0), P'(Y|A = 1)) \leq \delta$, and a prediction algorithm $f : \mathcal{X} \rightarrow \{0, 1\}$ has an error probability of $\epsilon$ w.r.t $P'$, i.e., $Pr\{f(X) \neq Y\} \leq \epsilon$, then we have:

$$\mathbb{TV}(P'(f(X)|A = 0), P'(f(X)|A = 1)) \leq \epsilon(1/2P'(A = 0) + 1/2P'(A = 1)) + \delta. \tag{2}$$

where $\mathbb{TV}$ is the total variation distance.

**Proposition 2.** If a randomized classifier $f$ has an expected error probability less than $\epsilon$, $\mathbb{E}_{X,Y \sim P'}(\mathbb{1}\{f(X) \neq Y\}) \leq \epsilon$, and $\mathbb{TV}(P, P') \leq \delta$, then the expected error probability of $f$ with respect to the distribution $P$ can be bounded as follows: $\mathbb{E}_{X,Y \sim P}(\mathbb{1}\{f(X) \neq Y\}) \leq \epsilon + 2\delta$.

**Proposition 3.** If a prediction algorithm $f : \mathcal{X} \rightarrow \{0, 1\}$ approximately satisfies SP, that is $\mathbb{TV}(P(f(X)|A = 0), P(f(X)|A = 1)) \leq \delta_1$ for a distribution $P'$ and we have $\mathbb{TV}(P, P') \leq \delta_2$, then we have:

$$\mathbb{TV}(P'(f(X)|A = 0), P'(f(X)|A = 1)) \leq \delta_2 h(p_0, p_1) + \delta_1, \tag{3}$$

where $p_0 = \min\{P(A = 0), P'(A = 0)\}$ and $p_1 = \min\{P(A = 1), P'(A = 1)\}$, and $h$ is a function we introduce in the proof.

Our definition here is in line with the definition of fairness for SDG in FairGAN [10] and [12], while it is different from the definition proposed in DECAF [11]. We have provided a discussion on the different definitions in the appendix.

## 3 Counterfactual fairness

In this section, we provide a method to create a dataset that is counterfactually fair. The definition of counterfactual fairness for a predictor was given in Section 2, which can be modified as follows for a SDG:

**Definition 4.** A generator producing samples $(X, A, Y)$ with distribution $P'$ is counterfactually fair if:

$$P'(Y_{A \leftarrow a} = y|X = x, A = a) = P'(Y_{A \leftarrow a'} = y|X = x, A = a), \tag{4}$$

for all $y \in \mathcal{Y}, x \in \mathcal{X}, a, a' \in \mathcal{A}$.

Here we should highlight that a method for SDG is proposed in [12] to satisfy counterfactual fairness. However, although they cite [9], and claim that they are attempting to satisfy this condition, they are actually considering interventions, not counterfactuals, and thus their method should be considered as an attempt to satisfy the following fairness constraint proposed in [15]:

**Definition 5.** (Discrimination avoiding through causal reasoning): A generator said to be fair if the following equation holds:

$$P(Y = y|X = x, do(A = a)) = P(Y = y|X = x, do(A = a')). \tag{5}$$

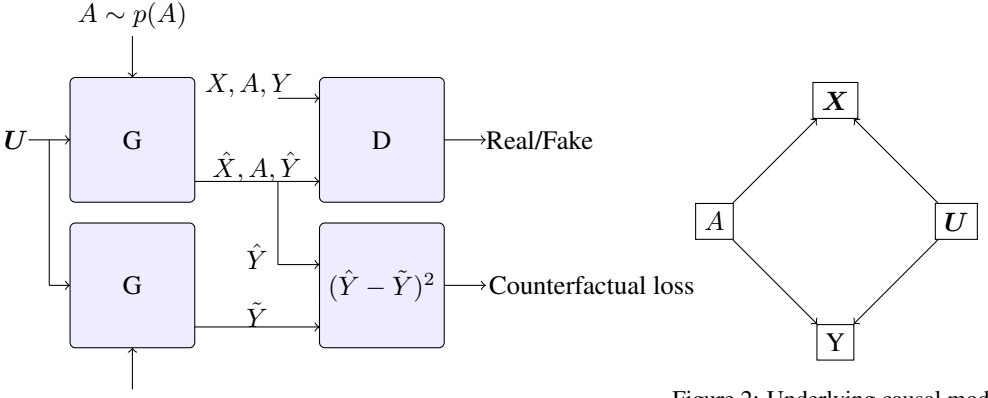

Figure 2: Underlying causal model

Figure 1: Counterfactual Fairness

This definition is based on the *do* operator (intervention) and thus is different from Definition 4, which considers counterfactuals. For more details about differences of these two regimes see supplementary material of [9] and [16][Chapter 4].

Consider the causal structure in Figure 2. Here we assume that we have the observed features $X$, sensitive attribute $A$, label $Y$, and an unobserved variable $U$. Following the example in [9], we can consider the law school admission problem, where $X$ is a three dimensional feature vector. The features include GPA, entrance exam score (LSAT), and gender. $A$ is race and $Y$ is the first year average grade (the output). Here $U$, the unobserved variable, is the true knowledge of law of the student. The proposed method is presented in Figure 1. We are using a conditional GAN structure, where $A$ is fed to the generator as the condition and the generator produces $X$ and $Y$, given the sensitive attribute, and random variable $U$. We are assuming here, that $U$ is the unobserved variable. This architecture is consistent with the casual model. Our goal is to force $G$ to produce the same output for the same individual (i.e., when $U$ is fixed) in the counterfactual world. Note that this is a stronger condition comparing to the Definition 4 (in Proposition 4, we show that this is a sufficient condition). Hence, we add an additional penalty term to the generator loss to enforce CF constraint. Let us denote the output of the generator by $G(u, a)$ where $u$ is an instance of random variable $U$ and $a$ is a sample of $A$. Also, we denote by $G_Y(u, a)$ the generated output $\hat{Y}$. Using a vanilla GAN our modified loss functions for the generator and the discriminator are as follows:

$$\mathcal{L}_D = \mathbb{E}_{x,a,y \sim P_{(X,A,Y)}} \log D(x, a, y) + \mathbb{E}_{u,a \sim P(U)P(A)}[\log(1 - D(G(u, a)))] \tag{6}$$

$$\mathcal{L}_G = -\mathbb{E}_{u,a \sim P(U)P(A)}[\log(1 - D(G(u, a)) - \lambda(G_Y(u, a) - G_Y(u, \neg a))^2] \tag{7}$$

The parameter $\lambda$ trades off the faithfulness to the underlying distribution and satisfying the fairness notion. Comparing (7) and (1), it can be observed that the first term of (7) attempts to minimise the distance between $P'$ and $P$ (when using vanilla GAN this will be Jensen-Shannon distance [17], note that in experiments we use Wasserstein distance which is known to be more stable [18]). The second term in (7), encourages the model to satisfy the fairness constraint in (1). For achieving counterfactual fairness, one sufficient condition that is proposed in [9] is to only use variables which are not descendants of $A$ in the casual graph, i.e., $U$ in our setting, to produce $Y$. Thus, the fact that we are using $A$ in our structure to create $Y$ might seem counter-intuitive. However, note that not using descendants of sensitive attributes is only a sufficient condition, and it is possible to achieve counterfactual fairness, while using these variables if the model cancels out their effect, and this is exactly what counterfactual loss does. If the counterfactual loss condition is perfectly satisfied then we have the following proposition.

**Proposition 4.** If the generator $G$ has zero counterfactual loss, i.e., $G_Y(u, a) = G_Y(u, \neg a)$ for all $u \in \mathcal{U}$ and $a \in \{0, 1\}$, then the produced distribution $P'(X, A, Y)$ satisfies the counterfactual fairness constraint given in (4).

*Proof.* See the appendix for the proof. □

## 3.1 Generalizing the causal model

Using the technique proposed in Section 5.1 of [11], our method can be generalized to any given causal model represented by a directed acyclic graph (DAG). Following Pearl's notation [19], consider a triple of $(U, V, F)$, where $V$ is all observed variables (in our setting $X$, $A$, and $Y$), $U$ represents all unobserved variables, and $F$ is a set of functions $\{f_1, \cdots, f_n\}$, that are corresponding to each $V_i \in V$, such that $V_i = f_i(pa_i, U_{pa_i})$, where $pa_i \subseteq V \setminus \{V_i\}$ and $U_{pa_i}$ are observable and unobservable parents of $V_i$. These equations are called structural equations. Now each of these equations can be modelled by a separate generator $G_i : \mathbb{R}^{|pa_i| + |U_{pa_i}|} \to \mathbb{R}$. Features are generated sequentially following the order imposed by the underlying causal graph. Assuming that parents of $V_i$ are already generated, we generate $\hat{V}_i = G_i(\hat{pa}_i, U_{pa_i})$, where $\hat{pa}_i$ are generated parents of $V_i$. The fact that underlying graph is acyclic, enables this ordering. We refer to Section 5.1 of [11] for a detailed explanation of this method. For imposing CF, we can penalize the generator similar to (7). The value of $A$ should first be flipped, while all unobserved values are fixed. Then we update the value of all descendants of $A$ including potentially $Y$, and compute the counterfactual loss. Note that if $Y$ is not a descendant of $A$, then counterfactual fairness will hold by definition. A more detailed discussion and an example are given in supplementary material for a general causal model.

**Remark.** The case where $A$ is not a binary variable can also be handled. Instead if flipping $A$ we can choose uniformly at random a value from $\mathcal{A} \setminus \{A\}$ and compute the counterfactual loss. Alternatively, we can compute $Y$ for all $A' \in \mathcal{A} \setminus \{A\}$ and define the penalty to be the average of counterfactual loss for all $A'$.

**Training differentially private fair models:** Note that it is possible to make our proposed model differentially private by using DP-SGD technique [20]. The differential privacy constraint is imposed by adding adequate noise (e.g. Gaussian noise) to the clipped gradients of the discriminator. Adding noise is only required for the discriminator since the true data is only fed to the discriminator, and the post-processing theorem [3] guarantees that there is no privacy leakage by the generator. This is true for all GAN based methods that we discussed, including FairGAN [10], DECAF [11], and CFGAN [12].

## 4 Experiments

**Law school experiment for counterfactual fairness:** Similar to [9], here we use the law school admission example to evaluate our counterfactually fair generator. The dataset is constructed via a survey conducted by The Law School Admission Council across 163 law schools in the United States [21]. It contains information on 21,790 students. We have access to the the following features: the entrance exam scores (LSAT), the grade-point average (GPA), the first year average grade (FYA), along with race and gender of students. Now the goal is to predict FYA as a proxy for the student's success using other features in the dataset. However, note that race and gender are sensitive attributes (here we only consider race as the sensitive attribute, as [9] found that the data is counterfactually fair w.r.t gender), thus the school wants the predictions not to be biased by these features. However, due to social factors, the other features, i.e., LSAT, GPA, and FYA may be biased by these attributes. Now the goal is to construct a counterfactually fair predictor of FYA.

Let us denote the real dataset by $D$. We split this dataset 80/20 into $D_{\text{train}}/D_{\text{test}}$ subsets. First we use $D_{\text{train}}$ to train the generative model proposed in Section 3 and produce the fair dataset $D'$ of the same size as $D_{\text{train}}$. Now we train a random forest regression model[3] on $D'$ (we have used a random forest as it significantly outperforms linear models), on the fair dataset, to estimate FYA using the rest of the features (i.e., LSAT, GPA, race, and gender). Denote this regression model with $f_{\text{fair}}$. We also train two baseline models using the real data $D$. $f_{\text{full}}$ is a random forest model trained on $D_{\text{train}}$ using all features (we expect this model to have the best accuracy and to be the most unfair model). Secondly, $f_{\text{unaware}}$ is another linear regression model using only GPA and LSAT (thus is unaware of sensitive attributes). The accuracy of these models in terms of RMSE on the test data ($D_{\text{test}}$) is reported in Table 1. We can see that there is a slight reduction in the accuracy when we use the synthetic data instead of the real data. Also, as we increase $\lambda$, the accuracy decreases because $\lambda$ controls the trade-off between the fairness level and $d(P, P')$. Larger $\lambda$ leads to larger distance between true distribution $P$ and generated distribution $P'$.

---

[3]We used the default model of sklearn package [22]

|        | $f_{\text{full}}$ | $f_{\text{unaware}}$ | $f_{\text{fair}}$ ($\lambda = 0.01$) | $f_{\text{fair}}$ ($\lambda = 0.1$) | $f_{\text{fair}}$ ($\lambda = 0.2$) |
|--------|-------|---------|---------------------|--------------------|--------------------|
| RMSE   | 0.256 | 0.257   | 0.259               | 0.261              | 0.263              |

Table 1: RMSE of fair and baseline models in terms of RMSE.

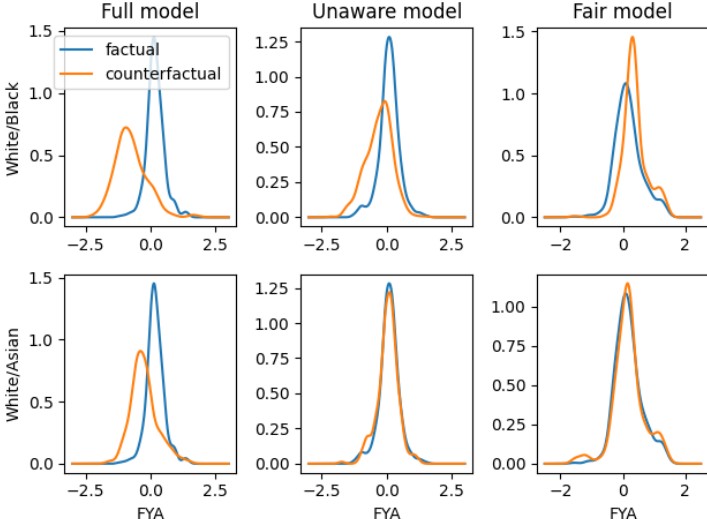

Figure 3: The distribution of estimated FYA for three models. For the fair model we chose $\lambda = 0.1$. The factual curve is representing White race, and counterfactual curve is Black, and Asian in row one and two respectively.

For assessing whether the learned algorithm $f_{\text{fair}}$ is counterfactually fair, the distribution of estimated FYA for both factual and counterfactual samples is required. To produce counterfactual samples, similar to [9], we fit the following linear model described in equation (8) as a counterfactual simulator on the data. Note that we are interested in satisfying the CF notion with respect to the true distribution, thus we use $D$ to train the simulator. This model follows the casual graph in Figure 2. In this example, $R$ is representing race, $S$ is gender, and $U$ is the unobserved variable representing the true knowledge of law. Similar to [9], we use an MCMC model to estimate coefficients in equation (8) and also to estimate the value of $U$ for each sample in the test data.

$$
\begin{aligned}
\text{GPA} &\sim \mathcal{N}(b_G + w_G^U U + w_G^R R + w_G^S S, \sigma_G), & \text{FYA} &\sim \mathcal{N}(w_F^U U + w_F^R R + w_F^S S, 1) \\
\text{LSAT} &\sim \text{Poisson}(\exp(b_L + w_L^U U + w_L^R R + w_L^S S)), & U &\sim \mathcal{N}(0, 1)
\end{aligned}
\tag{8}
$$

For a given sample in the test data, we use the model simulator to get the counterfactual samples and then feed both of these samples to $f_{\text{fair}}$ and also to two baseline methods. The distribution of estimated FYA for the factual and counterfactual samples are shown in Figure 3. We expect these two distributions to align if CF holds. In Figure 3, we consider two scenarios. In the first row, we compare White vs Black (White is factual and Black is counterfactual). It can be observed that the full model is not fair, Unaware model improves fairness, and the fair model is the most fair of the three. In the second row, we compare White and Asian. We can see that the unfairness is less for this experiment, and both unaware and fair model seem to satisfy the CF constraint for this pair of sensitive attributes.

## 5 Conclusion

In this paper, we formalised the definition of fairness for a synthetic data generator and showed why this definition is useful. We also proposed a new method for synthesising counterfactually fair data using GANs. There are several directions for future work. For example, here we considered statistical parity definition to prove propositions 1 to 3, and proving similar propositions for other fairness notions is one direction of future work. Another direction is proposing similar GAN-based methods for other fairness notions.

**Acknowledgment.** This work is partially supported by the NSF under grants IIS-2301599 and ECCS-2301601. MA and AE are supported in part by EPSRC via EP/V056883/1 at The Alan Turing Institute

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
