# OpenReview forum: "Counterfactual Fairness in Synthetic Data Generation"
_NeurIPS.cc/2022/Workshop/SyntheticData4ML — Neurips 2022 SyntheticData4ML_

### Official Review · Reviewer_zMUA · 2022-10-18
**A compelling but (at-times) hard to read paper**

**Rating:** 7
**Confidence:** 4

**Review:**

Paper summary: The authors articulate the problem of data scarcity in the ML field. They argue, like others before, that synthetic datasets can play a role in helping to amend this problem. The authors propose a fairness definition for synthetic data generation (SDG) that is separate from previous fair-ML definitions given it is not reliant on a model's prediction. They also propose a pre-processing type method that attempts to de-bias a dataset through the synthetic data generation process before a model is trained on the data. The goal of their method is to take out as much discriminatory bias as possible while also still resembling the real data. They also discuss differentially private SGD in the introduction and refer back to it when articulating how their method could be made differentially private which is a sound goal for a practitioner. The authors conduct a law school case study where they aim for counterfactual fairness and their SGD method proves quite effective--more effective than the other two models tested.

Thoughts: The quality of the research is sound but the writing quality could be improved. The originality is high and the authors do a great job in comparing how their work relates to and differs from other (substantial) papers and frameworks already published (e.g., FairGAN, DECAF, CFGAN). The significance of their work is strong since they present a fairness definition for SGD specifically and propose a method for actually reaching counterfactual fairness through SGD. Regarding clarity, much of the mathematical proofs are in the appendix which resulted in shortened explanations in the actual paper, leaving the reader wanting more information. I would recommend some of the arguments and explanations in the appendix be brought into the actual paper--this can be done by cutting down and making the writing more succinct. The conclusion (not an actual section in the paper) is tiny and does not refer to any future work or limitations which should be added. I would be curious to see if the authors plan to test other fairness definitions with their SGD method.

Misc.: Last sentence of intro edits: "Our contributions are as follows: formalising the definition…and proposing a new method…" Edits in 2.1: "The appropriate notion may be different for different datasets/tasks and for different policy makers." and "Here, we have a 'fair dataset' and no predictor immediately available; thus, …" Edits in 2.2: "Therefore, we argue that it is…"

Pros: quality of research, significance of contributions, knowledge of literature, proofs, experimental results

Cons: quality of writing and conciseness, too much was fit into the paper (and too much left for the appendix), lacking of future work and limitations discussed

---

### Official Review · Reviewer_2qqv · 2022-10-19
**Counterfactual Fairness in Synthetic Data Generation**

**Rating:** 6
**Confidence:** 3

**Review:**

The paper proposes a fairness mechanism during synthetic data generation. The problem is important and the current paper provides a counterfactual-based method to handle fair synthetic data generation. Please find feedback below:
1) The definition of fairness makes sense. However, the reviewer felt that fairness needs to link with biasness caused by A on critical features during the counter-factual optimization. The current formulation does not distinguish between X and sensitive features where bias is not acceptable.
2) The framing also will have a significant impact on the model performance. The output only captures & defined the distribution similarity. The author should also benchmark the 1) model performance, 2) Compare the impact on A & impact on critical feature importance. Wouldn't this will be the same as removing A from the modelling in the case of LSAT experimentation presented in the paper?
3) Author should compare the proposed framework with other counter-factual baselines such as "Fairness through Equality of Effort"

---

### Meta-Review · Area_Chair_kd16 · 2022-10-20

**Recommendation:** Accept